# Measurement-induced entanglement and teleportation on a noisy quantum processor

Google Quantum AI and Collaborators*

Measurement has a special role in quantum theory[1]: by collapsing the wavefunction, it can enable phenomena such as teleportation[2] and thereby alter the 'arrow of time' that constrains unitary evolution. When integrated in many-body dynamics, measurements can lead to emergent patterns of quantum information in space–time[3–10] that go beyond the established paradigms for characterizing phases, either in or out of equilibrium[11–13]. For present-day noisy intermediate-scale quantum (NISQ) processors[14], the experimental realization of such physics can be problematic because of hardware limitations and the stochastic nature of quantum measurement. Here we address these experimental challenges and study measurement-induced quantum information phases on up to 70 superconducting qubits. By leveraging the interchangeability of space and time, we use a duality mapping[9,15–17] to avoid mid-circuit measurement and access different manifestations of the underlying phases, from entanglement scaling[3,4] to measurement-induced teleportation[18]. We obtain finite-sized signatures of a phase transition with a decoding protocol that correlates the experimental measurement with classical simulation data. The phases display remarkably different sensitivity to noise, and we use this disparity to turn an inherent hardware limitation into a useful diagnostic. Our work demonstrates an approach to realizing measurement-induced physics at scales that are at the limits of current NISQ processors.

The stochastic, non-unitary nature of measurement is a foundational principle in quantum theory and stands in stark contrast to the deterministic, unitary evolution prescribed by Schrödinger's equation[1]. Because of these unique properties, measurement is key to some fundamental protocols in quantum information science, such as teleportation[2], error correction[19] and measurement-based computation[20]. All these protocols use quantum measurements, and classical processing of their outcomes, to build particular structures of quantum information in space–time. Remarkably, such structures may also emerge spontaneously from random sequences of unitary interactions and measurements. In particular, 'monitored' circuits, comprising both unitary gates and controlled projective measurements (Fig. 1a), were predicted to give rise to distinct non-equilibrium phases characterized by the structure of their entanglement[3,4,21–23], either 'volume law'[24] (extensive) or 'area law'[25] (limited), depending on the rate or strength of measurement.

In principle, quantum processors allow full control of both unitary evolution and projective measurements (Fig. 1a). However, despite their importance in quantum information science, the experimental study of measurement-induced entanglement phenomena[26,27] has been limited to small system sizes or efficiently simulatable Clifford gates. The stochastic nature of measurement means that the detection of such phenomena requires either the exponentially costly post-selection of measurement outcomes or more sophisticated data-processing techniques. This is because the phenomena are visible only in the properties of quantum trajectories; a naive averaging of experimental repetitions

incoherently mixes trajectories with different measurement outcomes and fully washes out the non-trivial physics. Furthermore, implementing the model in Fig. 1a requires mid-circuit measurements that are often problematic on superconducting processors because the time needed to perform a measurement is a much larger fraction of the typical coherence time than it is for two-qubit unitary operations. Here we use space–time duality mappings to avoid mid-circuit measurements, and we develop a diagnostic of the phases on the basis of a hybrid quantum-classical order parameter (similar to the cross-entropy benchmark in ref. 28) to overcome the problem of post-selection. The stability of these quantum information phases to noise is a matter of practical importance. Although relatively little is known about the effect of noise on monitored systems[29–31], noise is generally expected to destabilize measurement-induced non-equilibrium phases. Nonetheless, we show that noise serves as an independent probe of the phases at accessible system sizes. Leveraging these insights allows us to realize and diagnose measurement-induced phases of quantum information on system sizes of up to 70 qubits.

The space–time duality approach[9,15–17] enables more-experimentally convenient implementations of monitored circuits by leveraging the absence of causality in such dynamics. When conditioning on measurement outcomes, the arrow of time loses its unique role and becomes interchangeable with spatial dimensions, giving rise to a network of quantum information in space–time[32] that can be analysed in multiple ways. For example, we can map one-dimensional (1D) monitored circuits (Fig. 1a) to 2D shallow unitary circuits with measurements only

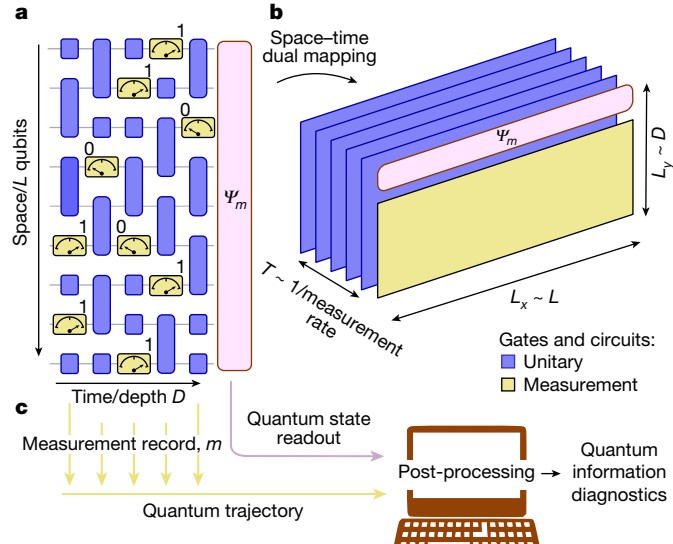

**Fig. 1 | Monitored circuits and space–time duality mapping. a**, A random (1 + 1)-dimensional monitored quantum circuit composed of both unitary gates and measurements. **b**, An equivalent dual (1 + 1)-dimensional shallow circuit of size $L_x \times L_y$ and depth $T$ with all measurements at the final time formed from a space–time duality mapping of the circuit in **a**. Because of the non-unitary nature of measurements, there is freedom as to which dimensions are viewed as 'time' and which as 'space'. In this example, $L_y$ is set by the (1 + 1)D circuit depth and $L_x$ by its spatial size, and $T$ is set by the measurement rate. **c**, Classical post-processing on a computer of the measurement record (quantum trajectory), and quantum-state readout of a monitored circuit can be used to diagnose the underlying information structures in the system.

at the final step[17] (Fig. 1b and Supplementary Information section 5), thereby addressing the experimental issue of mid-circuit measurement.

We began by focusing on a special class of 1D monitored circuits that can be mapped by space–time duality to 1D unitary circuits. These models are theoretically well understood[15,16] and are convenient to implement experimentally. For families of operations that are dual to unitary gates (Supplementary Information), the standard model of monitored dynamics[3,4] based on a brickwork circuit of unitary gates and measurements (Fig. 2a) can be equivalently implemented as a unitary circuit when the space and time directions are exchanged (Fig. 2b), leaving measurements only at the end. The desired output state $|\Psi_m\rangle$ is prepared on a temporal subsystem (in a fixed position at different times)[33]. It can be accessed without mid-circuit measurements by using ancillary qubits initialized in Bell pairs ($Q'_1...Q'_{12}$ in Fig. 2c) and SWAP gates, which teleport $|\Psi_m\rangle$ to the ancillary qubits at the end of the circuit (Fig. 2c). The resulting circuit still features post-selected measurements but their reduced number (relative to a generic model; Fig. 2a) makes it possible to obtain the entropy of larger systems, up to all 12 qubits ($Q'_1...Q'_{12}$), in individual quantum trajectories.

Previous studies[15,16] predicted distinct entanglement phases for $|\psi_m\rangle$ as a function of the choice of unitary gates in the dual circuit: volume-law entanglement if the gates induce an ergodic evolution, and logarithmic entanglement if they induce a localized evolution. We implemented unitary circuits that are representative of the two regimes, built from two-qubit fermionic simulation (fSim) unitary gates[34] with swap angle $\theta$ and phase angle $\phi = 2\theta$, followed by random single-qubit $Z$ rotations. We chose angles $\theta = 2\pi/5$ and $\theta = \pi/10$ because these are dual to non-unitary operations with different measurement strengths (Fig. 2d and Supplementary Information).

To measure the second Renyi entropy for qubits composing $|\Psi_m\rangle$, randomized measurements[35,36] are performed on $Q'_1...Q'_{12}$. Figure 2e shows the entanglement entropy as a function of subsystem size. The first gate set gives rise to a Page-like curve[24], with entanglement entropy growing linearly with subsystem size up to half the system and then ramping down. The second gate set, by contrast, shows a weak, sublinear dependence of entanglement with subsystem size. These findings are consistent with the theoretical expectation of distinct entanglement phases (volume-law and logarithmic, respectively) in monitored circuits that are space–time dual to ergodic and localized unitary circuits[15,16].

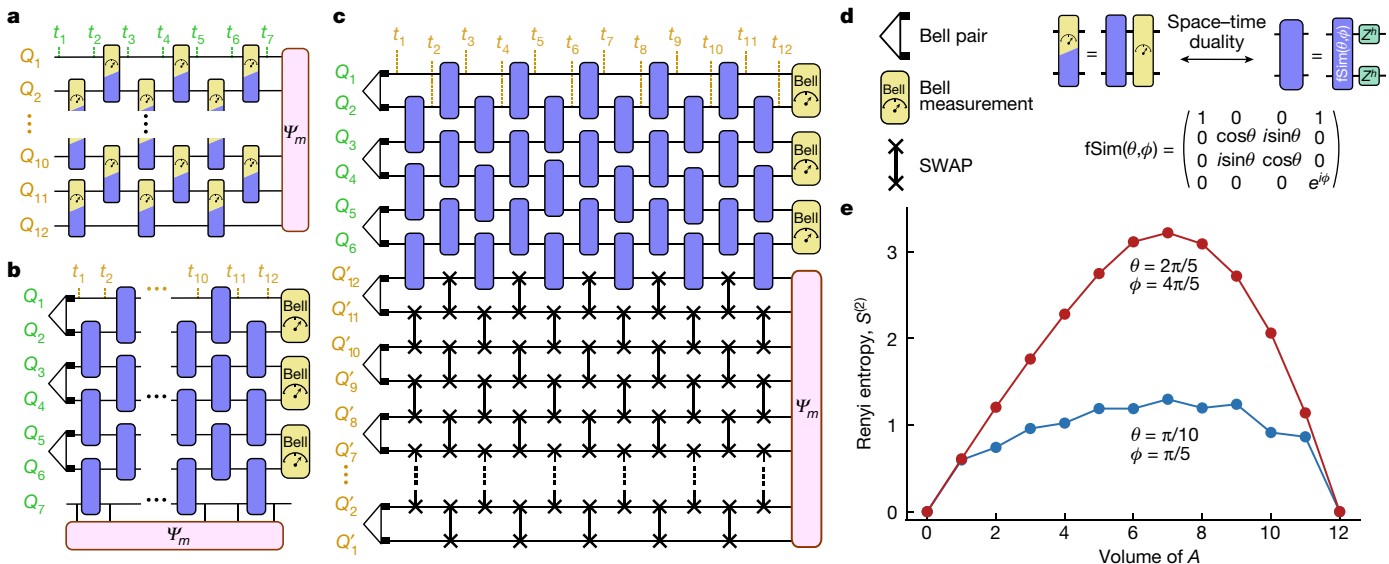

**Fig. 2 | Implementation of space–time duality in 1D. a**, A quantum circuit composed of non-unitary two-qubit operations in a brickwork pattern on a chain of 12 qubits with 7 time steps. Each two-qubit operation can be a combination of unitary operations and measurement. **b**, The space–time dual of the circuit shown in **a** with the roles of space and time interchanged. The 12-qubit wavefunction $|\Psi_m\rangle$ is temporally extended along $Q_7$. **c**, In the experiment on a quantum processor, a set of 12 ancillary qubits $Q'_1...Q'_{12}$ and a network of SWAP gates are used to teleport $|\Psi_m\rangle$ to the ancillary qubits. **d**, Illustration of the two-qubit gate composed of an fSim unitary and random $Z$ rotations with its space–time dual, which is composed of a mixture of unitary and measurement operations. The power $h$ of the $Z$ rotation is random for every qubit and periodic with each cycle of the circuit. **e**, Second Renyi entropy as a function of the volume of a subsystem $A$ from randomized measurements and post-selection on $Q_1...Q_6$. The data shown are noise mitigated by subtracting an entropy density matching the total system entropy. See the Supplementary Information for justification.

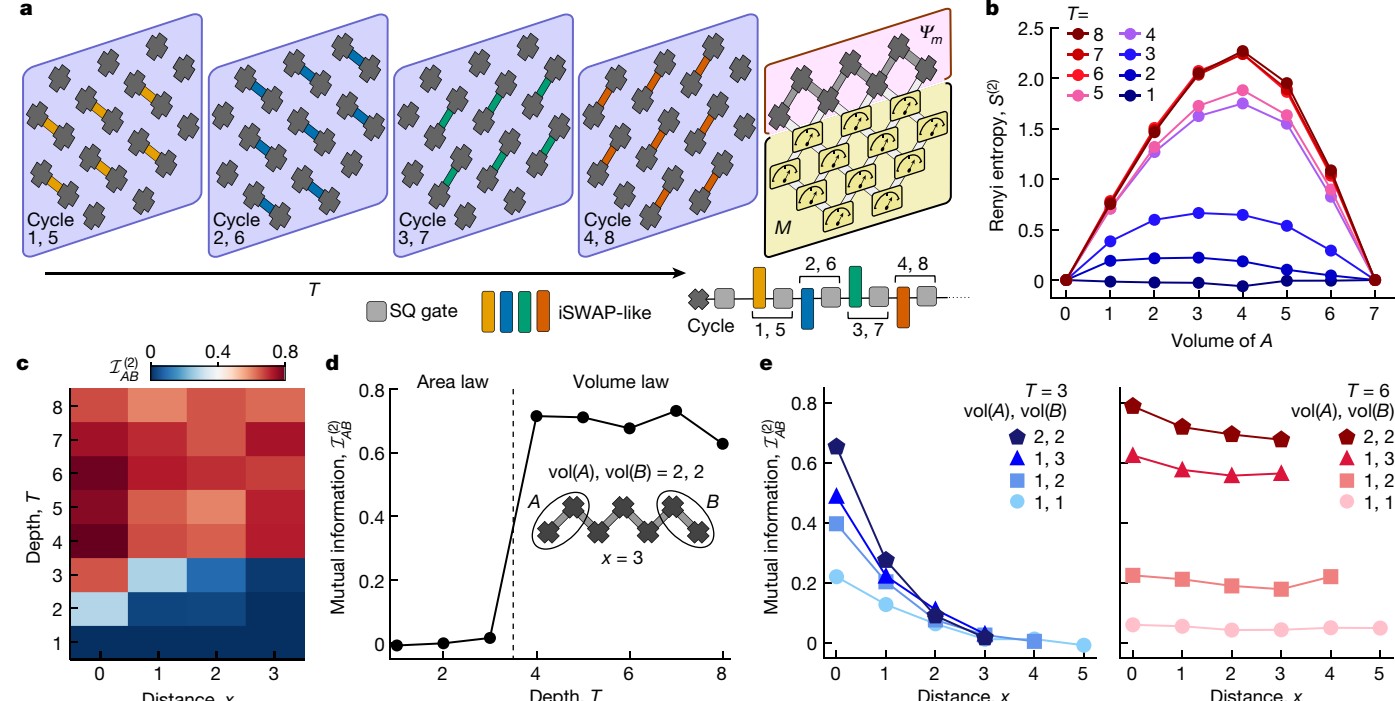

**Fig. 3 | 1D entanglement phases obtained from 2D shallow quantum circuits. a**, Schematic of the 2D grid of qubits. At each cycle (blue boxes) of the circuit, random single-qubit and two-qubit iSWAP-like gates are applied to each qubit in the cycle sequence shown. The random single-qubit gate (SQ, grey) is chosen randomly from the set $\left\{\sqrt{X^{\pm 1}}, \sqrt{Y^{\pm 1}}, \sqrt{W^{\pm 1}}, \sqrt{V^{\pm 1}}\right\}$, where $W = (X+Y)/\sqrt{2}$ and $V = (X-Y)/\sqrt{2}$. At the end of the circuit, the lower $M = 12$ qubits are measured and post-selected on the most probable bitstring. **b**, Second Renyi entropy of contiguous subsystems $A$ of the $L = 7$ edge qubits at various depths. The measurement is noise mitigated in the same way as in Fig. 2. **c**, Second Renyi mutual information $\mathcal{I}_{AB}^{(2)}$ between two-qubit subsystems $A$ and $B$ against depth $T$ and distance $x$ (the number of qubits between $A$ and $B$). **d**, $\mathcal{I}_{AB}^{(2)}$ as a function of $T$ for two-qubit subsystems $A$ and $B$ at maximum separation. **e**, $\mathcal{I}_{AB}^{(2)}$ versus $x$ for $T = 3$ and $T = 6$ for different volumes of $A$ and $B$.

A phase transition between the two can be achieved by tuning the $(\theta, \phi)$ fSim gate angles.

We next moved beyond this specific class of circuits with operations restricted to be dual to unitary gates, and instead investigated quantum information structures arising under more general conditions. Generic monitored circuits in 1D can be mapped onto shallow circuits in 2D, with final measurements on all but a 1D subsystem[17]. The effective measurement rate, $p$, is set by the depth of the shallow circuit, $T$, and the number of measured qubits, $M$. Heuristically, $p \approx M/(M+L)T$ (the number of measurements per unitary gate), where $L$ is the length of the chain of unmeasured qubits hosting the final state for which the entanglement structure is being investigated. Thus, for large $M$, a measurement-induced transition can be tuned by varying $T$. We ran 2D random quantum circuits[28] composed of iSWAP-like and random single-qubit rotation unitaries on a grid of 19 qubits (Fig. 3a), with $T$ varying from 1 to 8. For each depth, we post-selected on measurement outcomes of $M = 12$ qubits and left behind a 1D chain of $L = 7$ qubits; the entanglement entropy was then measured for contiguous subsystems $A$ by using randomized measurements. We observed two distinct behaviours over a range of $T$ values (Fig. 3b). For $T < 4$, the entropy scaling is subextensive with the size of the subsystem, whereas for $T \geq 4$, we observe an approximately linear scaling.

The spatial structure of quantum information can be further characterized by its signatures in correlations between disjointed subsystems of qubits: in the area-law phase, entanglement decays rapidly with distance[37], whereas in a volume-law phase, sufficiently large subsystems may be entangled arbitrarily far away. We studied the second Renyi mutual information

$$\mathcal{I}_{AB}^{(2)} = S_A^{(2)} + S_B^{(2)} - S_{AB}^{(2)}, \qquad (1)$$

between two subsystems $A$ and $B$ as a function of depth $T$, and the distance (the number of qubits) $x$ between them (Fig. 3c). For maximally separated subsystems $A$ and $B$ of two qubits each, $\mathcal{I}_{AB}^{(2)}$ remains finite for $T \geq 4$, but it decays to 0 for $T \leq 3$ (Fig. 3d). We also plotted $\mathcal{I}_{AB}^{(2)}$ for subsystems $A$ and $B$ with different sizes ($T = 3$ and $T = 6$) as a function of $x$ (Fig. 3e). For $T = 3$ we observed a rapid decay of $\mathcal{I}_{AB}^{(2)}$ with $x$, indicating that only nearby qubits share information. For $T = 6$, however, $\mathcal{I}_{AB}^{(2)}$ does not decay with distance.

The observed structures of entanglement and mutual information provide strong evidence for the realization of measurement-induced area-law ('disentangling') and volume-law ('entangling') phases. Our results indicate that there is a phase transition at critical depth $T \simeq 4$, which is consistent with previous numerical studies of similar models[17,18,38]. The same analysis without post-selection on the $M$ qubits (Supplementary Information) shows vanishingly small mutual information, indicating that long-ranged correlations are induced by the measurements.

The approaches we have followed so far are difficult to scale for system sizes greater than 10–20 qubits[27], owing to the exponentially increasing sampling complexity of post-selecting measurement outcomes and obtaining entanglement entropy of extensive subsystems of the desired output states. More scalable approaches have been recently proposed[39–42] and implemented in efficiently simulatable (Clifford) models[26]. The key idea is that diagnostics of the entanglement structure must make use of both the readout data from the quantum state $|\Psi_m\rangle$ and the classical measurement record $m$ in a classical post-processing step (Fig. 1c). Post-selection is the conceptually simplest instance of this idea: whether quantum readout data are accepted or rejected is conditional on $m$. However, because each instance of the experiment returns a random quantum trajectory[43] from $2^M$ possibilities

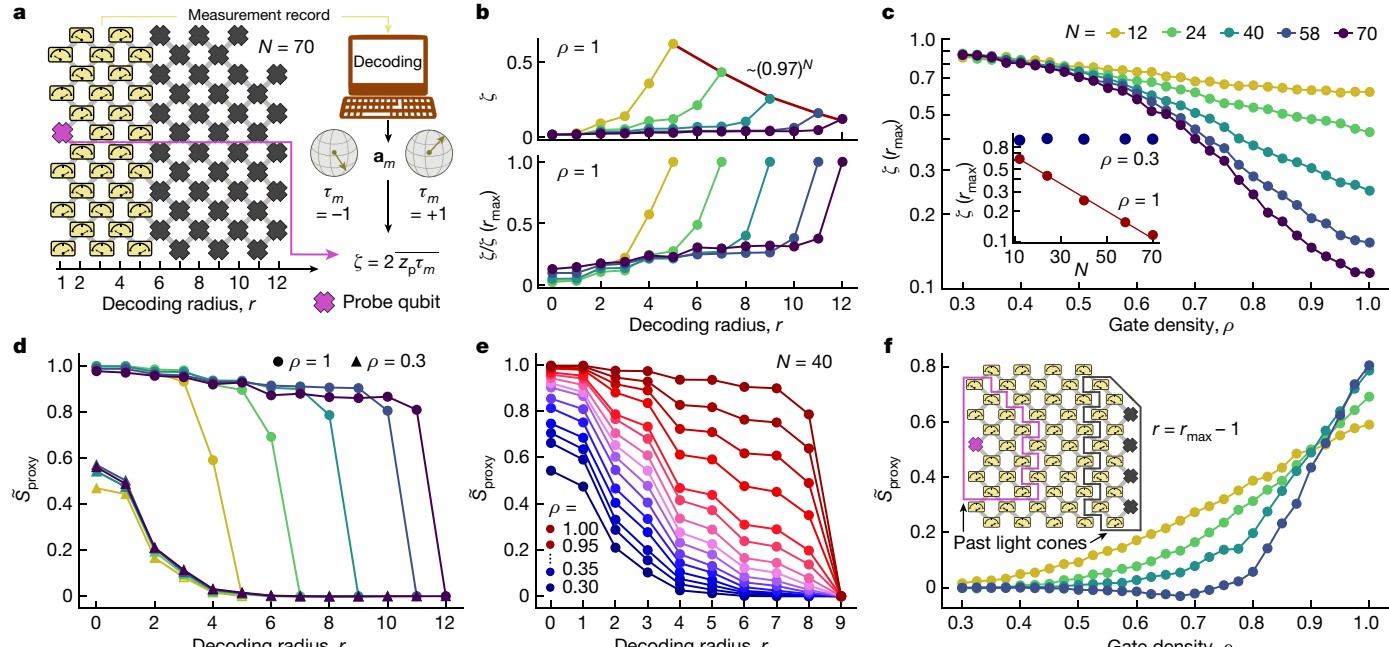

**Fig. 4 | Decoding of local order parameter, measurement-induced teleportation and finite-size analysis. a**, Schematic of the processor geometry and decoding procedure. The gate sequence is the same as in Fig. 3 with depth $T = 5$. The decoding procedure involves classically computing the Bloch vector $\mathbf{a}_m$ of the probe qubit (pink) conditional on the experimental measurement record $m$ (yellow). The order parameter $\zeta$ is calculated by means of the cross-correlation between the measured probe bit $z_p$ and $\tau_m = \mathrm{sign}(\mathbf{a}_m \cdot \hat{z})$, which is $+1$ if $\mathbf{a}_m$ points above the equator of the Bloch sphere and $-1$ if it points below. **b**, Decoded order parameter $\zeta$ and error-mitigated order parameter $\tilde{\zeta} = \zeta / \zeta(r_{max})$ as a function of the decoding radius $r$ for different $N$ and $\rho = 1$. **c**, $\zeta(r_{max})$ as a function of the gate density $\rho$ for different $N$. The inset shows that for small $\rho$, $\zeta(r_{max})$ remains constant as a function of $N$ (disentangling phase), whereas for larger $\rho$, $\zeta(r_{max})$ decays exponentially with $N$, implying sensitivity

to noise of arbitrarily distant qubits (entangling phase). **d**, Error-mitigated proxy entropy $\tilde{S}_{proxy}$ as a function of the decoding radius for $\rho = 0.3$ (triangles) and $\rho = 1$ (circles). In the disentangling phase, $\tilde{S}_{proxy}$ decays rapidly to 0, independent of the system size. In the entangling phase, $\tilde{S}_{proxy}$ remains large and finite up to $r_{max} - 1$. **e**, $\tilde{S}_{proxy}$ at $N = 40$ as a function of $r$ for different $\rho$, revealing a crossover between the entangling and disentangling phases for intermediate $\rho$. **f**, $\tilde{S}_{proxy}$ at $r = r_{max} - 1$ as a function of $\rho$ for $N = 12, 24, 40$ and 58 qubits. The curves for different sizes approximately cross at $\rho_c \approx 0.9$. Inset, schematic showing the decoding geometry for the experiment. The pink and grey lines encompass the past light cones (at depth $T = 5$) of the probe qubit and traced-out qubits at $r = r_{max} - 1$, respectively. Data were collected from 2,000 random circuit instances and 1,000 shots each for every value of $N$ and $\rho$.

(where $M$ is the number of measurements), this approach incurs an exponential sampling cost that limits it to small system sizes. Overcoming this problem will ultimately require more-sample-efficient strategies that use classical simulation[39,40,42], possibly followed by active feedback[39].

Here we have developed a decoding protocol that correlates quantum readout and the measurement record to build a hybrid quantum–classical order parameter for the phases that is applicable to generic circuits and does not require active feedback on the quantum processor. A key idea is that the entanglement of a single 'probe' qubit, conditioned on measurement outcomes, can serve as a proxy for the entanglement phase of the entire system[39]. This immediately eliminates one of the scalability problems: measuring the entropy of extensive subsystems. The other problem— post-selection—is removed by a classical simulation step that allows us to make use of all the experimental shots and is therefore sample efficient.

This protocol is illustrated in Fig. 4a. Each run of the circuit terminates with measurements that return binary outcomes $\pm 1$ for the probe qubit, $z_p$, and the surrounding $M$ qubits, $m$. The probe qubit is on the same footing as all the others and is chosen at the post-processing stage. For each run, we classically compute the Bloch vector of the probe qubit, conditional on the measurement record $m$, $\mathbf{a}_m$ (Supplementary Information). We then define $\tau_m = \mathrm{sign}(\mathbf{a}_m \cdot \hat{z})$, which is $+1$ if $\mathbf{a}_m$ points above the equator of the Bloch sphere, and $-1$ otherwise. The cross-correlator between $z_p$ and $\tau_m$, averaged over many runs of the experiment such

that the direction of $\mathbf{a}_m$ is randomized, yields an estimate of the length of the Bloch vector, $\zeta \simeq \overline{|\mathbf{a}_m|}$, which can in turn be used to define a proxy for the probe's entropy:

$$\zeta = 2\overline{z_p\,\tau_m}, \qquad S_{proxy} = -\log_2[(1 + \zeta^2)/2], \qquad (2)$$

where the overline denotes averaging over all the experimental shots and random circuit instances. A maximally entangled probe corresponds to $\zeta = 0$.

In the standard teleportation protocol[2], a correcting operation conditional on the measurement outcome must be applied to retrieve the teleported state. In our decoding protocol, $\tau_m$ has the role of the correcting operation, restricted to a classical bit-flip, and the cross-correlator describes the teleportation fidelity. In the circuits relevant to our experiment (depth $T = 5$ on $N \leq 70$ qubits), the classical simulation for decoding is tractable. For arbitrarily large circuits, however, the existence of efficient decoders remains an open problem[39,41,44]. Approximate decoders that work efficiently in only part of the phase diagram, or for special models, also exist[39], and we have implemented one such example based on matrix product states (Supplementary Information).

We applied this decoding method to 2D shallow circuits that act on various subsets of a 70-qubit processor, consisting of $N = 12, 24, 40, 58$ and 70 qubits in approximately square geometries (Supplementary Information). We chose a qubit near the middle of one side as the probe and computed the order parameter $\zeta$ by decoding measurement

outcomes up to $r$ lattice steps away from that side while tracing out all the others (Fig. 4a). We refer to $r$ as the decoding radius. Because of the measurements, the probe may remain entangled even when $r$ extends past its unitary light cone, corresponding to an emergent form of teleportation[18].

As seen in Fig. 3, the entanglement transition occurs as a function of depth $T$, with a critical depth $3 < T_c < 4$. Because $T$ is a discrete parameter, it cannot be tuned to finely resolve the transition. To do this, we fix $T = 5$ and instead tune the density of the gates, so each iSWAP-like gate acts with probability $\rho$ and is skipped otherwise, setting an 'effective depth' $T_{eff} = \rho T$; this can be tuned continuously across the transition. Results for $\zeta(r)$ at $\rho = 1$ (Fig. 4b) reveal a decay with system size $N$ of $\zeta(r_{max})$, where $r = r_{max}$ corresponds to measuring all the qubits apart from the probe. This decay is purely due to noise in the system.

Remarkably, sensitivity to noise can itself serve as an order parameter for the phase. In the disentangling phase, the probe is affected by noise only within a finite correlation length, whereas in the entangling phase it becomes sensitive to noise anywhere in the system. In Fig. 4c, $\zeta(r_{max})$ is shown as a function of $\rho$ for several $N$ values, indicating a transition at a critical gate density $\rho_c$ of around 0.6–0.8. At $\rho = 0.3$, which is well below the transition, $\zeta(r_{max})$ remains constant as $N$ increases (inset in Fig. 4c). By contrast, at $\rho = 1$ we fit $\zeta(r_{max})$ at around $0.97^N$, indicating an error rate of around 3% per qubit for the entire sequence. This is approximately consistent with our expectations for a depth $T = 5$ circuit based on individual gate and measurement error rates (Supplementary Information). This response to noise is analogous to the susceptibility of magnetic phases to a symmetry-breaking field[7,30,31,45] and therefore sharply distinguishes the phases only in the limit of infinitesimal noise. For finite noise, we expect the $N$ dependence to be cut off at a finite correlation length. We do not see the effects of this cut-off at system sizes accessible to our experiment.

As a complementary approach, the underlying behaviour in the absence of noise may be estimated by noise mitigation. To do this, we define the normalized order parameter $\tilde{\zeta}(r) = \zeta(r)/\zeta(r_{max})$ and proxy entropy $\tilde{S}_{proxy}(r) = -\log_2[(1 + \tilde{\zeta}(r)^2)/2]$. The persistence of entanglement with increasing $r$, corresponding to measurement-induced teleportation[18], indicates the entangling phase. Figure 4d shows the noise-mitigated entropy for $\rho = 0.3$ and $\rho = 1$, revealing a rapid, $N$-independent decay in the former and a plateau up to $r = r_{max} - 1$ in the latter. At fixed $N = 40$, $\tilde{S}_{proxy}(r)$ displays a crossover between the two behaviours for intermediate $\rho$ (Fig. 4e).

To resolve this crossover more clearly, we show $\tilde{S}_{proxy}(r_{max} - 1)$ as a function of $\rho$ for $N = 12–58$ (Fig. 4e). The accessible system sizes approximately cross at $\rho_c \approx 0.9$. There is an upward drift of the crossing points with increasing $N$, confirming the expected instability of the phases to noise in the infinite-system limit. Nonetheless, the signatures of the ideal finite-size crossing (estimated to be $\rho_c \simeq 0.72$ from the noiseless classical simulation; Supplementary Information) remain recognizable at the sizes and noise rates accessible in our experiment, although they are moved to larger $\rho_c$. A stable finite-size crossing would mean that the probe qubit remains robustly entangled with qubits on the opposite side of the system, even when $N$ increases. This is a hallmark of the teleporting phase[18], in which quantum information (aided by classical communication) travels faster than the limits imposed by the locality and causality of unitary dynamics. Indeed, without measurements, the probe qubit and the remaining unmeasured qubits are causally disconnected, with non-overlapping past light cones[46] (pink and grey lines in the inset in Fig. 4f).

Our work focuses on the essence of measurement-induced phases: the emergence of distinct quantum information structures in space–time. We used space–time duality mappings to circumvent mid-circuit measurements, devised scalable decoding schemes based on a local probe of entanglement, and used hardware noise to study these phases on up to 70 superconducting qubits. Our findings highlight the practical limitations of NISQ processors imposed by finite coherence.

By identifying exponential suppression of the decoded signal in the number of qubits, our results indicate that increasing the size of qubit arrays may not be beneficial without corresponding reductions in noise rates. At current error rates, extrapolation of our results (at $\rho = 1$, $T = 5$) to an $N$-qubit fidelity of less than 1% indicates that arrays of more than around 150 qubits would become too entangled with their environment for any signatures of the ideal (closed system) entanglement structure to be detectable in experiments. This indicates that there is an upper limit on qubit array sizes of about $12 \times 12$ for this type of experiment, beyond which improvements in system coherence are needed.

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

# Article

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

**Google Quantum AI and Collaborators**

J. C. Hoke[1,2,12], M. Ippoliti[2,3,12], E. Rosenberg[1,4], D. Abanin[1], R. Acharya[1], T. I. Andersen[1], M. Ansmann[1], F. Arute[1], K. Arya[1], A. Asfaw[1], J. Atalaya[1], J. C. Bardin[1,5], A. Bengtsson[1], G. Bortoli[1], A. Bourassa[1], J. Bovaird[1], L. Brill[1], M. Broughton[1], B. B. Buckley[1], D. A. Buell[1], T. Burger[1], B. Burkett[1], N. Bushnell[1], Z. Chen[1], B. Chiaro[1], D. Chik[1], J. Cogan[1], R. Collins[1], P. Conner[1], W. Courtney[1], A. L. Crook[1], B. Curtin[1], A. G. Dau[1], D. M. Debroy[1], A. Del Toro Barba[1], S. Demura[1], A. Di Paolo[1], I. K. Drozdov[1,6], A. Dunsworth[1], D. Eppens[1], C. Erickson[1], E. Farhi[1], R. Fatemi[1], V. S. Ferreira[1], L. F. Burgos[1], E. Forati[1], A. G. Fowler[1], B. Foxen[1], W. Giang[1], C. Gidney[1], D. Gilboa[1], M. Giustina[1], R. Gosula[1], J. A. Gross[1], S. Habegger[1], M. C. Hamilton[1,7], M. Hansen[1], M. P. Harrigan[1], S. D. Harrington[1], P. Heu[1], M. R. Hoffmann[1], S. Hong[1], T. Huang[1], A. Huff[1], W. J. Huggins[1], S. V. Isakov[1], J. Iveland[1], E. Jeffrey[1], Z. Jiang[1], C. Jones[1], P. Juhas[1], D. Kafri[1], K. Kechedzhi[1], T. Khattar[1], M. Khezri[1], M. Kieferová[1,8], S. Kim[1], A. Kitaev[1], P. V. Klimov[1], A. R. Klots[1], A. N. Korotkov[1,9], F. Kostritsa[1], J. M. Kreikebaum[1], D. Landhuis[1], P. Laptev[1], K.-M. Lau[1], L. Laws[1], J. Lee[1,10], K. W. Lee[1], Y. D. Lensky[1], B. J. Lester[1], A. T. Lill[1], W. Liu[1], A. Locharla[1], O. Martin[1], J. R. McClean[1], M. McEwen[1], K. C. Miao[1], A. Mieszala[1], S. Montazeri[1], A. Morvan[1], R. Movassagh[1], W. Mruczkiewicz[1], M. Neeley[1], C. Neill[1], A. Nersisyan[1], M. Newman[1], J. H. Ng[1], A. Nguyen[1], M. Nguyen[1], M. Y. Niu[1], T. E. O'Brien[1], S. Omonije[1], A. Opremcak[1], A. Petukhov[1], R. Potter[1], L. P. Pryadko[1,11], C. Quintana[1], C. Rocque[1], N. C. Rubin[1], N. Saei[1], D. Sank[1], K. Sankaragomathi[1], K. J. Satzinger[1], H. F. Schurkus[1], C. Schuster[1], M. J. Shearn[1], A. Shorter[1], N. Shutty[1], V. Shvarts[1], J. Skruzny[1], W. C. Smith[1], R. Somma[1], G. Sterling[1], D. Strain[1], M. Szalay[1], A. Torres[1], G. Vidal[1], B. Villalonga[1], C. V. Heidweiller[1], T. White[1], B. W. K. Woo[1], C. Xing[1], Z. J. Yao[1], P. Yeh[1], J. Yoo[1], G. Young[1], A. Zalcman[1], Y. Zhang[1], N. Zhu[1], N. Zobrist[1], H. Neven[1], R. Babbush[1], D. Bacon[1], S. Boixo[1], J. Hilton[1], E. Lucero[1], A. Megrant[1], J. Kelly[1], Y. Chen[1], V. Smelyanskiy[1], X. Mi[1], V. Khemani[2✉] & P. Roushan[1✉]

[1]Google Research, Mountain View, CA, USA. [2]Department of Physics, Stanford University, Stanford, CA, USA. [3]Department of Physics, University of Texas at Austin, Austin, TX, USA. [4]Department of Physics, Cornell University, Ithaca, NY, USA. [5]Department of Electrical and Computer Engineering, University of Massachusetts, Amherst, MA, USA. [6]Department of Physics, University of Connecticut, Storrs, CT, USA. [7]Department of Electrical and Computer Engineering, Auburn University, Auburn, AL, USA. [8]QSI, Faculty of Engineering & Information Technology, University of Technology Sydney, Sydney, New South Wales, Australia. [9]Department of Electrical and Computer Engineering, University of California, Riverside, CA, USA. [10]Department of Chemistry, Columbia University, New York, NY, USA. [11]Department of Physics and Astronomy, University of California, Riverside, CA, USA. [12]These authors contributed equally: J. C. Hoke, M. Ippoliti. ✉e-mail: vkhemani@stanford.edu; pedramr@google.com

## Data availability

The data that support the findings in this study are available at https://doi.org/10.5281/zenodo.7949563 (ref. 47).

## Code availability

The code that supports the findings of this study is available from the corresponding authors upon reasonable request.

47. Hoke, J. C. Quantum information phases in space-time: measurement-induced entanglement and teleportation on a noisy quantum processor. *Zenodo* https://doi.org/10.5281/zenodo.7949563 (2023).

**Acknowledgements** We acknowledge discussions with E. Altman, Y. Bao, M. Block, M. Gullans, Y. Li and L. Susskind. M.I. and V.K. thank T. Rakovszky for collaboration. D.B. is a CIFAR associate fellow in the Quantum Information Science Program. M.I. is supported in part by the Gordon and Betty Moore Foundation's EPiQS Initiative through grant GBMF8686. V.K. acknowledges support from the US Department of Energy, Office of Science, Basic Energy Sciences under Early Career Award DE-SC0021111, the Alfred P. Sloan Foundation through a Sloan Research Fellowship and the Packard Foundation through a Packard Fellowship in Science and Engineering. Numerical simulations were performed in part using the Sherlock cluster at the Stanford Research Computing Center. We acknowledge the hospitality of the Kavli Institute for Theoretical Physics at the University of California, Santa Barbara (supported by NSF grant PHY-1748958).

**Author contributions** J.C.H., M.I., X.M., V.K. and P.R. designed the experiment, discussed the project, interpreted the results and wrote the manuscript. J.C.H. and M.I. wrote the Supplementary Information. J.C.H. implemented the experiment. J.C.H. and E.R. collected experimental data. M.I. designed the decoding protocol and implemented the classical simulations. X.M., V.K. and P.R. led and coordinated the project. Infrastructure support was provided by Google Quantum AI. All authors contributed to revising the manuscript and the Supplementary Information.

**Competing interests** The authors declare no competing interests.

**Additional information**
**Correspondence and requests for materials** should be addressed to V. Khemani or P. Roushan.
