## [Peer Review File · Nature]

Manuscript Title: Measurement-induced entanglement and teleportation on a noisy quantum processor

Reviewer Comments & Author Rebuttals

Reviewer Reports on the Initial Version:

Referees' comments:

Referee #1 (Remarks to the Author):

Dear Dr. Levi,

In the present manuscript, the authors experimentally study entanglement transitions in random monitored circuits. The last few years have seen a boom in the theoretical study of the emergent phenomena arising from quantum dynamics combining unitary gates and projective measurements. More recently, several quantum devices (including ions and superconducting qubits) have confirmed some of these predictions. The present experimental work presents a threefold innovation: (i) it implements previously-proposed spacetime dualities to study the entanglement transition without requiring the costly technology of mid-circuit measurements; (ii) using this trick, it pushes the study of this entanglement transition to considerably larger system sizes; and (iii) it uses the noise of the device to help detect the transition.

While any of the above three innovations might (in isolation) not meet the bar of a Nature publication, the combination of all three makes for a compelling work that could be a nice fit for Nature. Although the concept of spacetime duality to study monitored circuits is not new, I am not aware of a previous implementation. Moreover, its implementation on such remarkably large superconducting qubit chips (~ 70 qubits) makes this feat all the more impressive. On the conceptual side, I feel the main contribution of the present manuscript is the observation that the intrinsic noise of the device (which is usually seen as a nuisance) can be used as a probe to detect the transition. If the latter stands up to scrutiny, it provides a compelling advance in the field of entanglement transitions.

However, before I can recommend publication, I have a few important questions for the authors:

1) On p5, the authors note that "sensitivity to noise can itself serve as an order parameter", which seems to be nicely illustrated in Fig 4c, where ζ either approaches a nonzero constant ($\rho=0.3$) or decays with system size ($\rho=1$). However, on p6 the authors acknowledge that "the phases are ultimately unstable to noise in the infinite-system limit". These two statements are incompatible. The latter quote suggests that the former discussion is incomplete at best, if not simply wrong. If noise washes out the transition, then this implies that for any ρ (even $\rho=0.3$), ζ should eventually decay to zero (for large enough N), is this correct? If so, it becomes unclear how one uses noise as a probe. Since the conceptual message of using noise as a probe is one of the key parts of this work (as also emphasized in the abstract), the authors should clear up this point.

2) I had a hard time following the definition of ζ on p4, or at least why it should be a good indicator for $|\langle \vec{a}_m \rangle|$. Suppose for sake of discussion that the probe qubit was unentangled (i.e., $|\langle \vec{a}_m \rangle| = 1$), but suppose it points close to the equator (with a tiny positive component along the z-axis). Then $\tau_m = 1$. However, z_P will still give a small value and thus seemingly a small ζ , despite the probe qubit not being entangled. I tried to read the part of the SI on this topic, but it was rather lengthy and technical. In the SI, it is explained that in the Knapp reference,

\vec{a}_m is rotated to be along the z-axis. It is then stated that this is not the route followed by the present work, but there I could find a simple explanation of what is done instead (without having to wade through the technical details). It would be great if this part can be further clarified.

3) I could not find a statement about data availability. Have the authors made the data publicly available?

Referee #2 (Remarks to the Author):

Dear Editor,

The manuscript by Google Quantum AI and collaborators provides state-of-the-art insights into measurement-induced phases in currently available quantum processors.

In this report, I will summarize the main novelties and contributions of the paper after briefly reviewing its general context.

I will argue that the paper is of outstanding quality and a breakthrough in the timely field of monitored systems. For these reasons, I believe the manuscript should be published in Nature after the authors consider the comments and implement the minor revisions listed at the end of this report.

General context

In recent years, monitored quantum systems have been the focus of extensive studies due to their unconventional non-unitary effects stemming from the competition of the system dynamics and local measurements. The hallmark phenomenon is the rise of novel dynamical phases separated by measurement-induced phase transitions (MIPT).

In the simplistic case of interacting quantum circuits, the MIPTs differentiate an error-correcting phase (characterized by a volume-law scaling of the entanglement entropy) and an error-fragile (Zeno) phase with sub-extensive scaling of the entanglement entropy.

While theoretically, these frameworks have been copiously explored with well-defined phase diagrams, experimentally monitored systems are challenging to tame.

Indeed, measurement-induced phases are visible only in non-linear observables of the density matrix, such as the state's purity or entanglement content. (En passant, this point justifies the labeling "Quantum information phases" chosen by the authors).

Therefore, the observables conditioned onto the measurement results are required for these phenomena. This fact leads to the post-selection problem: since measurements are intrinsically stochastic, each trajectory occurs with probability 2^{-M} , with M the total number of measurements, resulting in an exponential overhead for the experiments.

Before this work, Ref. [27] tackled the post-selection problem via a brute-force experiment.

Instead, Ref. [26] combined decoding protocols based on their Clifford implementation to avoid post-selection. Still, the post-selection limitation remains open for generic systems and experimentally, despite critical theoretical insights in quantum-classical hybrid schemes (e.g., Ref. [34-36]).

Beyond the post-selection, mentioning two other sources for the experimental implementation of measurement-induced phases is essential. First, the noise due to decoherence affects any current quantum processors (justifying the name noisy intermediate scale quantum devices, NISQ).

Second, the physical implementation of these gates depends on the framework and is subject to specific problems.

For instance, for the superconducting, transmon-based circuits considered in this manuscript, the mid-circuit measurements require significant times compared to the coherence time of the unitary operations. Mid-circuit measures, therefore, constitute a general problem in the development of

measurement-induced phases.

Additionally, the gates are constructed via physical interactions between the lowest energy levels of their superconducting elements. In turn, they are subject to leakage effects arising from the interactions between low-lying and higher-excited states.

Main Novelties

This work resolves or mitigates the previously mentioned limitations in the experimental detection of measurement-induced phases.

i) First, the authors leverage space-time duality to avoid mid-circuit measurements while overcoming the previous limitations on the leakage errors on two-body unitary gates via a better choice of adiabatic interaction between the transmons.

ii) Second, the authors implement quantum-classical hybrid order parameters inspired by recent theoretical achievements (Ref. [34-36]).

iii) (Third) In this context, the hardware noise becomes a "feature" and can be used as a specific probe of the phases.

Notably, the gates considered are non-Clifford (i.e., they lead to non-stabilizer/magic states), allowing a much broader discussion than the setup in Ref. [26].

These contributions are demonstrated combinedly or separately in three setups, pronouncing the three parts of the manuscript.

1) Initially, the authors consider a 1D monitored circuit that maps to a 1D unitary circuit via space-time duality (see supplemental information (SI) S4).

In this case, the authors consider an error-mitigation method (SI, S3) to remove noisy contribution and estimate the entanglement entropy of the system. Furthermore, the choice of circuits also allows mitigation of the post-selection, requiring $O(\sqrt{M})$ compared to the M measurements of the brute-force approach (e.g., Ref. [26]). In turn, larger system sizes are amenable for this class of circuits.

2) Afterward, the authors illustrate how 2D shallow circuits with measurements on all but a 1D subsystem can be mapped to monitored circuits in 1+1D. The effective rate of measurements depends on the shallow depth T . Using the mutual information, the authors distinguish a volume and area law phase, respectively, at large and low T . (Error-mitigation applies as in SI S3, cf. Fig. 3).

3) Finally, using a decoding local order parameter based on classical-quantum hybrid simulations, the authors demonstrate how measurement-induced transitions can be scaled beyond the full post-selection limit (~ 20 qubits).

To do so, they consider a 2D shallow circuit over 70 qubits that can be efficiently simulated with classical methods and use a single ancilla qubit as a probe for the transition. Without noise, a transition occurs in the state's decodability, as pointed out in SI S6.

The authors reveal that a similar phenomenology occurs when noise is present: the decodability behavior is qualitatively captured in the experiment cf. Fig 4.

The main difference with the ideal simulation is a shift of the critical rate, which is expected as a weak noise quantitatively affects the phase diagram.

Recommendation

Given the discussion above over context and novelties, I recommend the manuscript for publication in Nature.

Said that I recommend the authors revise the Supplemental Information, which can be polished and clarified in the initial sections, and adjust a few minor suggestions over the Main Text.

Minor criticism, comments, and suggested changes

1) If I understand correctly, an important source of error in the quantum-classical hybrid order parameter is that it requires classical simulations. However, the noise in the actual device would lead to potentially uncontrolled errors compared to the ideal simulations. From the manuscript, this should not be the case for the setup considered by the authors. Is this fact related to the choice of local probe combined with the quantum-classical order parameter? Or are there other justifications for this uncontrolled error's (qualitative) irrelevancy in their setup? In any case, the authors should present a more detailed discussion about this point in the manuscript.

2) In S1, defining the characteristic relaxation time T_1 would be instructive. Similarly, it is not specified what the two-qubit Pauli error rates are (See also S2,c). Please insert a brief definition/characterization.

3) Eq. S1 is wrong. The element in the third row should be $[0, -i e^{i(\Delta_+ - \Delta_{\text{off}})} \sin \theta, e^{i(\Delta_+ - \Delta_-)} \cos \theta, 0]$. If I understand correctly, the SWAP gate is implemented adiabatically via the trapezoid function, and leads to $S(\theta, \Delta_+) = \exp [i \theta (e^{i \Delta_+} c_0^\dagger c_1 + \text{h.c.})]$. (Here, c_α and c^\dagger_α are the annihilation, creation fermionic operators. I denote $n_\alpha = c_\alpha c^\dagger_\alpha$). The conditional angle arise as a rotation $O = \exp [i \phi n_0 n_1]$, and the single phases are given in terms of f_0 and f_1 via $v_\alpha = \exp [i f_\alpha n_\alpha]$. In particular Δ_- and Δ_{off} are linear transformations of f_0 and f_1 . Is the above correct? I suggest the authors add a few details about these gates. Additionally, it would be helpful to reveal the relationship between the parameters $f_0, f_1, g_{\text{max}}, t_p, t_{\text{rise}}$ and the parameters of fSim.

4) I suggest the authors add a summary of the goals and strategy of section S2 or rephrase its subsections. There is some logical back-and-forth between subsections S2a-S2c, that renders the text difficult to read. Either the authors summarize the key aspects of the section and the logical flow at the beginning of S2, or they should polish the body text of S2.

5) The authors should define $P(s)$ explicitly in S3, and clarify they notation. This minor change (probably requiring two-three extra lines) would improve the manuscript's readability.

6) The authors should define explicitly what is random-Z SQ gate in Fig. 2c, main.

Referee #3 (Remarks to the Author):

In this work, the authors explored the possibility of using Google's quantum computer to experimentally study the measurement-induced phenomenon, which is an important step towards investigating measurement-induced physics on the current NISQ processors.

Incorporating measurement in unitary quantum circuits has become a new approach to generate many interesting quantum phases. However, most of these studies are limited to theoretical work because experimentally realizing these quantum phenomena is still quite challenging. The authors identified two technical problems that hinder experimental studies: (1) The coherence time limitation. Measurement gates typically take longer to apply than unitary gates, making it difficult to observe repeated mid-circuit measurements on current NISQ processors due to coherence time limitations, and (2) The post-selection problem, where obtaining the same quantum state multiple times in stochastic quantum dynamics can be quite challenging due to the exponential growth in the number of quantum trajectories under monitored quantum dynamics.

To address these challenges, the authors explored two setups: (1) The 1+1d unitary circuit, which is dual to a non-unitary circuit after exchanging the spatial and temporal directions, and (2) The 2d shallow circuit, which generated a 2d quantum state from a shallow circuit. By measuring a fraction of the bulk qubits, the authors explored the interesting entanglement structure of the post-measurement state. In both setups, the authors were able to observe measurement-induced phenomena on the current NISQ processors by using error mitigation, post-selection, and classical decoding algorithms.

This work demonstrates a new direction for introducing measurement gates to quantum circuits to generate new quantum phases on NISQ devices, despite the fragility of these phenomena. Given the lack of experimental progress on monitored quantum dynamics on NISQ, this work fills up this gap and may have significant implications for quantum computing and quantum information science.

Based on these findings, I believe this work holds great promise for publication in Nature. However, before reaching a decision, I recommend that the referee address the following issues:

The authors present some experimental data of the entanglement entropy (along the temporal direction) for 1+1d unitary circuits that are equivalent to non-unitary dynamics when the spatial and temporal directions are exchanged. However, it appears that only a small subset of non-unitary gates can be mapped to pure unitary gates using this approach. The majority of non-unitary dynamics remains non-unitary after exchanging the spatial and temporal directions, which limits the applicability of this technique to investigate measurement-induced physics. It would be beneficial if the authors could address this limitation in the paper.

This paper examines both the 1+1d unitary circuit and the 2d shallow circuit. However, the connection between the two should be clarified as the authors repeatedly emphasize that they can be understood using the space-time duality approach. I disagree with this statement as I believe they are fundamentally different. The 1+1d unitary circuit is simply a tool for understanding a limited subset of non-unitary circuits, while the 2d shallow circuit provides a new avenue for exploring measurement-induced physics. While the 1+1d unitary circuit can be understood in terms of space-time duality, the 2d shallow circuit is a distinct model. By measuring the bulk qubits of the quantum state, the remaining unmeasured qubits can exhibit interesting entanglement structures.

In comparison to the 1+1d unitary circuit, the 1d state generated from the 2d shallow circuit (following bulk measurement) appears to be more intriguing. It appears that the 2d shallow circuit approach can reproduce numerous 1d phases generated by the 1+1d monitored dynamics. As a result, I recommend that the authors concentrate on the 2d shallow circuit model and potentially remove some of the discussion about the 1+1d unitary circuit or even eliminate it altogether.

Author Rebuttals to Initial Comments:

Referees' comments:

Referee #1 (Remarks to the Author):

Dear Dr. Levi,

In the present manuscript, the authors experimentally study entanglement transitions in random monitored circuits. The last few years have seen a boom in the theoretical study of the emergent phenomena arising from quantum dynamics combining unitary gates and projective measurements. More recently, several quantum devices (including ions and superconducting qubits) have confirmed some of these predictions. The present experimental work presents a threefold innovation: (i) it implements previously-proposed spacetime dualities to study the entanglement transition without requiring the costly technology of mid-circuit measurements; (ii) using this trick, it pushes the study of this entanglement transition to considerably larger system sizes; and (iii) it uses the noise of the device to help detect the transition.

While any of the above three innovations might (in isolation) not meet the bar of a Nature publication, the combination of all three makes for a compelling work that could be a nice fit for Nature. Although the concept of spacetime duality to study monitored circuits is not new, I am not aware of a previous implementation. Moreover, its implementation on such remarkably large superconducting qubit chips (~ 70 qubits) makes this feat all the more impressive. On the conceptual side, I feel the main contribution of the present manuscript is the observation that the intrinsic noise of the device (which is usually seen as a nuisance) can be used as a probe to detect the transition. If the latter stands up to scrutiny, it provides a compelling advance in the field of entanglement transitions.

We thank the reviewer for their review and positive comments regarding our manuscript, in particular highlighting the significance of identifying noise as an indicator of the transition.

However, before I can recommend publication, I have a few important questions for the authors:

1) On p5, the authors note that "sensitivity to noise can itself serve as an order parameter", which seems to be nicely illustrated in Fig 4c, where ζ either approaches a nonzero constant ($\rho=0.3$) or decays with system size ($\rho=1$). However, on p6 the authors acknowledge that "the phases are ultimately unstable to noise in the infinite-system limit". These two statements are incompatible. The latter quote suggests that the former discussion is incomplete at best, if not simply wrong. If noise washes out the transition, then this implies that for any ρ (even $\rho=0.3$), ζ should eventually decay to zero (for large enough N), is this correct? If so, it becomes unclear how one uses noise as a probe. Since the conceptual message of using noise as a probe is one of the key parts of this work (as also emphasized in the abstract), the authors should clear up this point.

The Referee points out an important and subtle issue. The resolution to this apparent contradiction is that *infinitesimal* noise distinguishes the two phases up to the thermodynamic (large- N) limit, while noise with finite strength ϵ introduces a finite length scale $\xi(\epsilon)$ (divergent as a power of $1/\epsilon$) that cuts off the N -dependence at a finite size.

It may be helpful to draw an analogy to more familiar phases of matter, e.g. ferromagnets. The ordered phase of a magnet spontaneously breaks rotational symmetry. This can be diagnosed by a divergence in the susceptibility to an applied magnetic field. At the same time, turning on a *finite* magnetic field explicitly breaks the rotational symmetry and eliminates the sharp distinction between the paramagnet and ferromagnet. In this analogy, the response to noise as $\sim f^N$ is the diverging susceptibility of the ordered phase. In the presence of finite noise strength ϵ , the N -dependence would still appear as a finite-size effect, but would eventually stop once the linear size of the system L is comparable to the noise-induced length scale $\xi(\epsilon)$. As the experimental data of Fig.4c shows, this condition is not met at the system sizes and noise strength of our work.

We illustrate this point with the help of numerical simulations. To make the effect visible, one can either increase system size or decrease the length scale $\xi(\epsilon)$, which is done by increasing noise strength. We use the second approach. We take systems of $N = 12$ and 24 qubits (same circuit realizations as in the experiment) and add noise of strength ϵ modeled by uncorrelated measurement error. The results for ζ as a function of gate density ρ are shown below for different noise strengths. We see that $\epsilon=0.02$ looks qualitatively similar to the experiment, as expected; however, as noise strength increases, the size dependence becomes weaker.

We expect this qualitative behavior would emerge in our Fig.4c if extended to much larger system sizes (hundreds of qubits). To clarify the text, we have added a sentence at the end of the paragraph on ‘sensitivity to noise’ (highlighted).

2) I had a hard time following the definition of zeta on p4, or at least why it should be a good indicator for $|\text{vec } a_m|$. Suppose for sake of discussion that the probe qubit was unentangled (i.e., $|\text{vec } a_m| = 1$), but suppose it points close to the equator (with a tiny positive component along the z-axis). Then $\tau_m = 1$. However, z_P will still give a small value and thus seemingly a small zeta, despite the probe qubit not being entangled. I tried to read the part of the SI on this topic, but it was rather lengthy and technical. In the SI, it is explained that in the Knapp reference, $\text{vec } a_m$ is rotated to be along the z-axis. It is then stated that this is not the route followed by the present work, but there I could find a simple explanation of what is done instead (without having to wade through the technical details). It would be great if this part can be further clarified.

We agree that in individual circuit instances, the value of zeta is not necessarily a good proxy for the length of the Bloch vector. The solution is in the fact that, across random circuit instances, the direction of the vector is randomized on the unit sphere, so that all three components of the Bloch vector have the same statistical distribution, and measuring any one of them is sufficient. In particular, as we discuss in Section VI.B of the SI, the average of $|a_m^z|$ (z-component of the vector) is one half of its average length, $|a_m|$. This is the reason why we define zeta with a factor of 2 up front (Eq.(2) of the main text), so that it cancels out the factor of $\frac{1}{2}$.

We have added the words “such that the direction of a_m is randomized” in the main text above Eq.(2), and a short explanation of this fact at the end of Section VI.A of the SI, so that the main idea can be conveyed separately from the technical details of Section VI.B. Both additions are highlighted.

3) I could not find a statement about data availability. Have the authors made the data publicly available?

We have added a data availability statement and the data can be found publicly available at: <https://doi.org/10.5281/zenodo.7949563>.

Referee #2 (Remarks to the Author):

Dear Editor,

The manuscript by Google Quantum AI and collaborators provides state-of-the-art insights into measurement-induced phases in currently available quantum processors. In this report, I will summarize the

main novelties and contributions of the paper after briefly reviewing its general context. I will argue that the paper is of outstanding quality and a breakthrough in the timely field of monitored systems. For these reasons, I believe the manuscript should be published in Nature after the authors consider the comments and implement the minor revisions listed at the end of this report.

We are excited to hear the reviewer's positive remarks recognizing the significance of our work. We thank the reviewer for reviewing our manuscript and address the reviewer's comments individually below.

General context

In recent years, monitored quantum systems have been the focus of extensive studies due to their unconventional non-unitary effects stemming from the competition of the system dynamics and local measurements. The hallmark phenomenon is the rise of novel dynamical phases separated by measurement-induced phase transitions (MIPT). In the simplistic case of interacting quantum circuits, the MIPTs differentiate an error-correcting phase (characterized by a volume-law scaling of the entanglement entropy) and an error-fragile (Zeno) phase with sub-extensive scaling of the entanglement entropy.

While theoretically, these frameworks have been copiously explored with well-defined phase diagrams, experimentally monitored systems are challenging to tame. Indeed, measurement-induced phases are visible only in non-linear observables of the density matrix, such as the state's purity or entanglement content. (En passant, this point justifies the labeling "Quantum information phases" chosen by the authors). Therefore, the observables conditioned onto the measurement results are required for these phenomena. This fact leads to the post-selection problem: since measurements are intrinsically stochastic, each trajectory occurs with probability 2^{-M} , with M the total number of measurements, resulting in an exponential overhead for the experiments.

Before this work, Ref. [27] tackled the post-selection problem via a brute-force experiment. Instead, Ref. [26] combined decoding protocols based on their Clifford implementation to avoid post-selection. Still, the post-selection limitation remains open for generic systems and experimentally, despite critical theoretical insights in quantum-classical hybrid schemes (e.g., Ref. [34-36]).

Beyond the post-selection, mentioning two other sources for the experimental implementation of measurement-induced phases is essential. First, the noise due to decoherence affects any current quantum processors (justifying the name noisy intermediate scale quantum devices, NISQ). Second, the physical implementation of these gates depends on the framework and is subject to specific problems. For instance, for the superconducting, transmon-based circuits considered in this manuscript, the mid-circuit measurements require significant times compared to the coherence time of the unitary operations. Mid-circuit measures, therefore, constitute a general problem in the development of measurement-induced phases.

Additionally, the gates are constructed via physical interactions between the lowest energy levels of their superconducting elements. In turn, they are subject to leakage effects arising from the interactions between low-lying and higher-excited states.

Main Novelties

This work resolves or mitigates the previously mentioned limitations in the experimental detection of measurement-induced phases.

i) First, the authors leverage space-time duality to avoid mid-circuit measurements while overcoming the previous limitations on the leakage errors on two-body unitary gates via a better choice of adiabatic interaction between the transmons.

ii) Second, the authors implement quantum-classical hybrid order parameters inspired by recent theoretical achievements (Ref. [34-36]).

iii) (Third) In this context, the hardware noise becomes a "feature" and can be used as a specific probe of the phases. Notably, the gates considered are non-Clifford (i.e., they lead to non-stabilizer/magic states), allowing a much broader discussion than the setup in Ref. [26].

These contributions are demonstrated combinedly or separately in three setups, pronouncing the three parts of the manuscript.

1) Initially, the authors consider a 1D monitored circuit that maps to a 1D unitary circuit via space-time duality (see supplemental information (SI) S4). In this case, the authors consider an error-mitigation method (SI, S3) to remove noisy contribution and estimate the entanglement entropy of the system. Furthermore, the choice of circuits also allows mitigation of the post-selection, requiring $O(\sqrt{M})$ compared to the M measurements of the brute-force approach (e.g., Ref. [26]). In turn, larger system sizes are amenable for this class of circuits.

2) Afterward, the authors illustrate how 2D shallow circuits with measurements on all but a 1D subsystem can be mapped to monitored circuits in 1+1D. The effective rate of measurements depends on the shallow depth T . Using the mutual information, the authors distinguish a volume and area law phase, respectively, at large and low T . (Error-mitigation applies as in SI S3, cf. Fig. 3).

3) Finally, using a decoding local order parameter based on classical-quantum hybrid simulations, the authors demonstrate how measurement-induced transitions can be scaled beyond the full post-selection limit (~ 20 qubits). To do so, they consider a 2D shallow circuit over 70 qubits that can be efficiently simulated with classical methods and use a single ancilla qubit as a probe for the transition. Without noise, a transition occurs in the state's decodability, as pointed out in SI S6. The authors reveal that a similar phenomenology occurs when noise is present: the decodability behavior is qualitatively captured in the experiment cf. Fig 4. The main difference with the ideal simulation is a shift of the critical rate, which is expected as a weak noise quantitatively affects the phase diagram.

Recommendation

Given the discussion above over context and novelties, I recommend the manuscript for publication in Nature. Said that I recommend the authors revise the Supplemental Information, which can be polished and clarified in the initial sections, and adjust a few minor suggestions over the Main Text.

Minor criticism, comments, and suggested changes

1) If I understand correctly, an important source of error in the quantum-classical hybrid order parameter is that it requires classical simulations. However, the noise in the actual device would lead to potentially uncontrolled errors compared to the ideal simulations. From the manuscript, this should not be the case for the setup considered by the authors. Is this fact related to the choice of local probe combined with the quantum-classical order parameter? Or are there other justifications for this uncontrolled error's (qualitative) irrelevancy in their setup? In any case, the authors should present a more detailed discussion about this point in the manuscript.

The classical simulations used to decode the experimental data and compute the quantum-classical order parameter are "exact", in the sense that they don't rely on any truncation. [For completeness we have also studied an approximate method, Sec.VII of the SI, which is not exact and introduces additional error, but this is not used in the main text.] The error comes from the quantum hardware, or rather from the mismatch between the real, noisy quantum evolution and the ideal model that is simulated. While it is possible to include a form of

error mitigation (e.g. by the division shown in Fig.4b), the error is ultimately uncontrolled, in the sense that the sharp distinction between phases would disappear in the large-system limit, at fixed noise rate. In other words, the finite error rate sets an upper limit for the size at which the transition can be resolved, as we discuss also in the reply to Referee 1. Nonetheless, for the system sizes accessible in experiment, evidence of the two phases is clearly visible in two complementary ways:

- From the susceptibility to noise, Fig.4c
- From noise-mitigated data, Fig.4f

We already addressed the fate of the second diagnostic under noise below the discussion of Fig.4f. We now also briefly discuss the same issue for the first diagnostic, in response to the question by Referee 1. We hope that this will be sufficient to clarify the discussion of this point. The changes have been highlighted in the main text.

2) In S1, defining the characteristic relaxation time T1 would be instructive. Similarly, it is not specified what the two-qubit Pauli error rates are (See also S2,c). Please insert a brief definition/characterization.

We have added additional details that T1 is found by running simultaneous population decay experiments on each qubit (i.e. measuring the decay rate of a qubit from the 1 state to the 0 state). We have also included that the two-qubit error rates are found via cross entropy benchmarking (XEB) and cited Arute *et al.* Nature (2019), where more specific details can be found. The changes have been highlighted in the supplement.

3) Eq. S1 is wrong. The element in the third row should be $[0, -i e^{i(\Delta_+ - \Delta_{-, \text{off}})} \sin \theta, e^{i(\Delta_+ - \Delta_-)} \cos \theta, 0]$.

If I understand correctly, the SWAP gate is implemented adiabatically via the trapezoid function, and leads to $S(\theta, \Delta_+) = \exp [i \theta (e^{i \Delta_+} c_0^\dagger c_1 + \text{h.c.})]$. (Here, c_α and c^\dagger_α are the annihilation, creation fermionic operators. I denote $n_\alpha = c_\alpha c^\dagger_\alpha$.) The conditional angle arise as a rotation $O = \exp [i \phi n_0 n_1]$, and the single phases are given in terms of f_0 and f_1 via $v_\alpha = \exp [i f_\alpha n_\alpha]$. In particular Δ_- and $\Delta_{-, \text{off}}$ are linear transformations of f_0 and f_1 .

Is the above correct? I suggest the authors add a few details about these gates. Additionally, it would be helpful to reveal the relationship between the parameters $f_0, f_1, g_{\text{max}}, t_p, t_{\text{rise}}$ and the parameters of fSim.

The typo in Eq. S1 has been fixed. We have added additional details in the section indicating how the fSim parameters depend on the pulse parameters: *“In terms of the coupler pulse parameters, both θ and ϕ scale linearly with t_p while $\theta \propto g_{\text{max}}$ and $\phi \propto g_{\text{max}}^2$. This difference in scaling enables independent control of θ and ϕ .”* Further details can be found in the supplementary information of Morvan *et al.* Nature 2022 (Ref. 2), which first introduced this fSim framework. The changes have been highlighted.

4) I suggest the authors add a summary of the goals and strategy of section S2 or rephrase its subsections. There is some logical back-and-forth between subsections S2a-S2c, that renders the text difficult to read. Either the authors summarize the key aspects of the section and the logical flow at the beginning of S2, or they should polish the body text of S2.

We agree with the reviewer that there is some level of back and forth in these sections. We have revised the text in these sections and added a summary of goals at the beginning to reflect the flow of logic: *“In the following sections we describe the implementation of arbitrary 2-qubit fSim gates with superconducting qubits. We then characterize the leakage errors that arise from these interactions. Finally, we discuss the specifics*

related to the calibration of the different fSim gates used in this experiment and their resulting fidelities.” This modification has been highlighted in the supplement.

5) The authors should define $P(s)$ explicitly in S3, and clarify their notation. This minor change (probably requiring two-three extra lines) would improve the manuscript's readability.

We have added additional text to explicitly state that $P(s)$ refers to the probability of measuring bitstring s .

6) The authors should define explicitly what is random-Z SQ gate in Fig. 2c, main.

We have redone this experiment and remade the figure (fig2.) to reflect this change. Random Z SQ gates are now the only SQ gates used and they are explicitly defined as the Pauli operator Z^h , where h is random.

Referee #3 (Remarks to the Author):

In this work, the authors explored the possibility of using Google's quantum computer to experimentally study the measurement induced phenomenon, which is an important step towards investigating measurement-induced physics on the current NISQ processors.

Incorporating measurement in unitary quantum circuits has become a new approach to generate many interesting quantum phases. However, most of these studies are limited to theoretical work because experimentally realizing these quantum phenomena is still quite challenging. The authors identified two technical problems that hinder experimental studies: (1) The coherence time limitation. Measurement gates typically take longer to apply than unitary gates, making it difficult to observe repeated mid-circuit measurements on current NISQ processors due to coherence time limitations, and (2) The post-selection problem, where obtaining the same quantum state multiple times in stochastic quantum dynamics can be quite challenging due to the exponential growth in the number of quantum trajectories under monitored quantum dynamics.

To address these challenges, the authors explored two setups: (1) The 1+1d unitary circuit, which is dual to a non-unitary circuit after exchanging the spatial and temporal directions, and (2) The 2d shallow circuit, which generated a 2d quantum state from a shallow circuit. By measuring a fraction of the bulk qubits, the authors explored the interesting entanglement structure of the post-measurement state. In both setups, the authors were able to observe measurement-induced phenomena on the current NISQ processors by using error mitigation, post-selection, and classical decoding algorithms.

This work demonstrates a new direction for introducing measurement gates to quantum circuits to generate new quantum phases on NISQ devices, despite the fragility of these phenomena. Given the lack of experimental progress on monitored quantum dynamics on NISQ, this work fills up this gap and may have significant implications for quantum computing and quantum information science.

Based on these findings, I believe this work holds great promise for publication in Nature. However, before reaching a decision, I recommend that the referee address the following issues:

We thank the reviewer for taking the time to review our work and their positive review of it. We address the issues they raised below.

The authors present some experimental data of the entanglement entropy (along the temporal direction) for 1+1d unitary circuits that are equivalent to non-unitary dynamics when the spatial and temporal directions are exchanged. However, it appears that only a small subset of non-unitary gates can be mapped to pure unitary gates using this approach. The majority of non-unitary dynamics remains non-unitary after exchanging the spatial and temporal directions, which limits the applicability of this technique to investigate measurement-induced physics. It would be beneficial if the authors could address this limitation in the paper.

We thank the reviewer for their comment and highlighting this issue. In the revised manuscript, we have made the introduction shorter and more focused on the main structure of the paper: in studying measurement induced phases on our hardware, there are two challenges: mid-circuit measurement and post-selection. Space-time duality (Fig 1 and 2 and 3) solves the mid-circuit measurement issue and to solve the post-selection we developed the hybrid classical-quantum method (Fig. 4).

We agree that the 1D models we consider in Fig2 only give access to a fine-tuned subset of monitored circuits. We have modified the main text to make this point more explicit and the passage is highlighted:

We begin by focusing on a special class of 1D monitored circuits that can be mapped by space-time duality to 1D unitary circuits. These models are theoretically well-understood and convenient to implement experimentally. For families of operations that are dual to unitary gates (see SI), the standard model of monitored dynamics based on a brickwork circuit of unitary gates and measurements (Fig. 2a) can be equivalently implemented as a unitary circuit upon exchanging the space and time directions (Fig. 2b), leaving measurements only at the end.

We indeed wanted to be clear that this is not a general-purpose solution but rather a special setting where experimental implementation is particularly convenient. This setting allows us to produce states with distinct entanglement structures on 12-qubit chains, among the largest entropy measurements carried out to date. Moreover there is theoretical evidence (see Ref.[16]) that the volume-law phase in these models is the same phase as that of “standard” monitored circuits (with e.g. the same nonthermal subleading corrections to entropy), so even though a small subset of circuits can be realized, these are enough to realize part of the general phase diagram.

This paper examines both the 1+1d unitary circuit and the 2d shallow circuit. However, the connection between the two should be clarified as the authors repeatedly emphasize that they can be understood using the space-time duality approach. I disagree with this statement as I believe they are fundamentally different. The 1+1d unitary circuit is simply a tool for understanding a limited subset of non-unitary circuits, while the 2d shallow circuit provides a new avenue for exploring measurement-induced physics. While the 1+1d unitary circuit can be understood in terms of space-time duality, the 2d shallow circuit is a distinct model. By measuring the bulk qubits of the quantum state, the remaining unmeasured qubits can exhibit interesting entanglement structures.

We agree with the referee that we needed to better clarify and distinguish between Fig. 2 and 3. The opening paragraphs of each part have been modified and highlighted to reflect the distinction.

We now make it clear that the 2D unitary shallow circuit model comes about after mapping from a 1+1D monitored circuit (Section V in the SI). This is the point of view taken in Ref. [17], which introduces the idea of “space evolution” for 2D shallow circuits. It is also arguably the more natural perspective when it comes to the “teleportation” data in Fig.4, with decoding radius playing the role of “time”.

While “space-time duality” has been used before to refer specifically to the 1D setup, in this work we use it as a general idea about mapping unitary evolutions to non-unitary ones by rearranging spatial and temporal dimensions of the dynamics, illustrated in Fig.1. This definition includes both the 1D and 2D cases: in the 1D case, a unitary circuit of dimensions (L,T) maps onto a non-unitary one of dimensions (T,L) ; in the 2D case, a unitary circuit of dimensions (L_x,L_y,T) maps onto a non-unitary one of dimensions (L_x*T, L_y) , see Sec. V in the SI. The microscopic implementation of the transformations is indeed different in the two cases, with significant consequences for the phase diagrams that can be explored (the 2D case is more general) and for the actual number of measurements needed in the implementation (the 1D case is more convenient). However we believe that the two are conceptually similar and belong together as experimental approaches towards the study of quantum information phases.

In comparison to the 1+1d unitary circuit, the 1d state generated from the 2d shallow circuit (following bulk measurement) appears to be more intriguing. It appears that the 2d shallow circuit approach can reproduce numerous 1d phases generated by the 1+1d monitored dynamics. As a result, I recommend that the authors concentrate on the 2d shallow circuit model and potentially remove some of the discussion about the 1+1d unitary circuit or even eliminate it altogether.

We agree with the Referee that the 2D shallow circuits yield a richer and more interesting model, which is the main focus of the work with two dedicated figures (Fig 3 and 4). In the process of revising the manuscript we have shortened the discussion of the 1D model and slightly reduced the size of Fig. 2. Nonetheless, we think the 1D model in Fig 2, despite its restriction to fine-tuned gate sets, is valuable as a setting to illustrate the mechanism of the space-time transformation, and to enable entropy measurements on larger systems (12 qubits). We also think it conceptually belongs in the paper as part of a unified approach to the study of quantum information phases. We hope that the referee agrees with us in keeping it in the paper now that the discussion about it is reduced.

Reviewer Reports on the First Revision:

Referees' comments:

Referee #1 (Remarks to the Author):

Dear Dr. Levi,

I thank the authors for their careful responses to my questions, and for the updated manuscript. The authors satisfactorily explain how the inherent noise can both be used as a probe of their transition while at the same time ultimately destroying the transition. In fact, I found their explanation---by making an analogy to the more familiar case of a ferromagnet---so useful that I encourage the authors to also briefly comment on this in the manuscript.

In light of their updated explanation about the effect of noise, my only remaining hesitation is that now their discussion of Fig 4f seems to slightly over-state their findings. While I am quite convinced and pleased by the authors discussion of panel c (in the context of noise detecting the transition), panels d through f consider a more uncontrolled noise-mitigated diagnostic. The authors suggest that panel f shows a (finite-size) crossing, albeit at a shifted ρ_c . I feel it would be less misleading to say that there is a drift with system size, as one can already observe in their largest system sizes shown (which is $N=58$ as $N=70$ is curiously not shown in this panel). Correspondingly, I also find the statement in the main text "Above this crossing, the probe qubit remains robustly entangled with qubits on the opposite side of the system even as N increases" prone to misinterpretation, as there is an N -dependent drift and I do not think the authors genuinely mean to claim it being 'robustly entangled'.

Despite the above remaining concerns, I would simply request that the editor asks the authors to carefully and seriously consider these points and to add a few explanatory notes in the manuscript. It does not affect the main findings of this work, and I would thus recommend this manuscript for publication in Nature. Indeed, this work represents an impressive combination of experimental accomplishments and some new conceptual insights, and I can imagine the demonstrated capabilities inspiring future works.

Referee #2 (Remarks to the Author):

Dear Editor,

The authors have resolved all the issues raised in the Referees' reports. The revised text includes important clarifications and additional context highlighting the relevance of the Authors' findings. Also, the revised Supplemental Information is more transparent and clear, raising the work's readability.

Given these considerations and the comments in my previous report, I suggest this manuscript for publication in Nature.

Referee #3 (Remarks to the Author):

The authors have addressed all of my questions listed in the report letter. The current version can be published on Nature.

Author Rebuttals to First Revision:**Referees' comments:**

Referee #1 (Remarks to the Author):

Dear Dr. Levi,

I thank the authors for their careful responses to my questions, and for the updated manuscript. The authors satisfactorily explain how the inherent noise can both be used as a probe of their transition while at the same time ultimately destroying the transition. In fact, I found their explanation---by making an analogy to the more familiar case of a ferromagnet---so useful that I encourage the authors to also briefly comment on this in the manuscript.

In light of their updated explanation about the effect of noise, my only remaining hesitation is that now their discussion of Fig 4f seems to slightly over-state their findings. While I am quite convinced and pleased by the authors discussion of panel c (in the context of noise detecting the transition), panels d through f consider a more uncontrolled noise-mitigated diagnostic. The authors suggest that panel f shows a (finite-size) crossing, albeit at a shifted ρ_c . I feel it would be less misleading to say that there is a drift with system size, as one can already observe in their largest system sizes shown (which is $N=58$ as $N=70$ is curiously not shown in this panel). Correspondingly, I also find the statement in the main text "Above this crossing, the probe qubit remains robustly entangled with qubits on the opposite side of the system even as N increases" prone to misinterpretation, as there is an N -dependent drift and I do not think the authors genuinely mean to claim it being 'robustly entangled'.

Despite the above remaining concerns, I would simply request that the editor asks the authors to carefully and seriously consider these points and to add a few explanatory notes in the manuscript. It does not affect the main findings of this work, and I would thus recommend this manuscript for publication in Nature. Indeed, this work represents an impressive combination of experimental accomplishments and some new conceptual insights, and I can imagine the demonstrated capabilities inspiring future works.

In responses to these additional helpful suggestions we have made small modifications to the text in the following ways:

1) We have added the following text:

“This response to noise is analogous to the susceptibility of magnetic phases to a symmetry-breaking field and thus sharply distinguishes the phases only in the limit of infinitesimal noise. For finite noise, we expect the N -dependence to be cut off at a finite correlation length.”

This replaces the previous text:

“This N -dependence is expected to be cut off by a noise-induced correlation length which diverges as noise strength goes to zero.”

2) We have also added the following text:

“The accessible system sizes approximately cross at $\rho_c \approx 0.9$. An upward drift of the crossing points with increasing N is visible, confirming the expected instability of the phases to noise in the infinite-system limit. Nonetheless, signatures of the ideal finite-size crossing (estimated to be $\rho_c \simeq 0.72$ from noiseless classical simulation, see SI) remain recognizable at the sizes and noise rate accessible in our experiment, although moved to larger ρ_c . A stable finite-size crossing would mean that the probe qubit remains robustly entangled with qubits on the opposite side of the system even as N increases.”

This replaces these previous two lines of text:

“The curves display a finite-size crossing at $\rho_c \approx 0.9$. Above this crossing, the probe qubit remains robustly entangled with qubits on the opposite side of the system even as N increases.”

“Noiseless classical simulation yields a crossing at $\rho_c \simeq 0.72$ (see SI). While the phases are ultimately unstable to noise in the infinite-system limit, the finite-size crossing remains recognizable at the sizes accessible in our experiment, although moved to larger ρ_c .”

Referee #2 (Remarks to the Author):

Dear Editor,

The authors have resolved all the issues raised in the Referees' reports. The revised text includes important clarifications and additional context highlighting the relevance of the Authors' findings. Also, the revised Supplemental Information is more transparent and clear, raising the work's

readability. Given these considerations and the comments in my previous report, I suggest this manuscript for publication in Nature.

Referee #3 (Remarks to the Author):

The authors have addressed all of my questions listed in the report letter. The current version can be published on Nature.